# Factors influencing subclinical atherosclerosis in patients with biopsy-proven nonalcoholic fatty liver disease

Taeang Arai[1]◉, Masanori Atsukawa 📷[1]◉*, Akihito Tsubota[2]◉, Tadamichi Kawano[1], Mai Koeda[3], Yuji Yoshida[3], Tomohide Tanabe[1], Tomomi Okubo[3], Korenobu Hayama[1], Ai Iwashita[1], Norio Itokawa[3], Chisa Kondo[1], Keiko Kaneko[1], Chiaki Kawamoto[1], Tsutomu Hatori[4], Naoya Emoto[5], Etsuko Iio[6], Yasuhito Tanaka[6], Katsuhiko Iwakiri[1]

**1** Division of Gastroenterology and Hepatology, Nippon Medical School, Tokyo, Japan, **2** Core Research Facilities for Basic Science, Research Center for Medical Sciences, The Jikei University School of Medicine, Tokyo, Japan, **3** Division of Gastroenterology, Nippon Medical School Chiba Hokusoh Hospital, Inzai, Japan, **4** Division of Pathology, Nippon Medical School Chiba Hokusoh Hospital, Inzai, Japan, **5** Division of Endocrinology, Nippon Medical School Chiba Hokusoh Hospital, Inzai, Japan, **6** Department of Virology and Liver Unit, Nagoya City University Graduate School of Medicinal Sciences, Nagoya, Japan

◉ These authors contributed equally to this work.
* momogachi@yahoo.co.jp

**Data Availability Statement:** All relevant data are within the manuscript and its Supporting Information files.

## Abstract

Although the presence of nonalcoholic fatty liver disease (NAFLD) is known to be related to subclinical atherosclerosis, the relationship between the severity of NAFLD and subclinical atherosclerosis is not clear. This study aimed to clarify the factors related to subclinical arteriosclerosis, including the histopathological severity of the disease and *PNPLA3* gene polymorphisms, in NAFLD patients. We measured brachial-ankle pulse wave velocity (baPWV) as an index of arterial stiffness in 153 biopsy-proven NAFLD patients. The baPWV values were significantly higher in the advanced fibrosis group than in the less advanced group (median, 1679 cm/s *vs* 1489 cm/s; $p = 5.49 \times 10^{-4}$). Multiple logistic regression analysis revealed that older age ($\geq$55 years) ($p = 8.57 \times 10^{-3}$; OR = 3.03), hypertension ($p = 1.05 \times 10^{-3}$; OR = 3.46), and advanced fibrosis ($p = 9.22 \times 10^{-3}$; OR = 2.94) were independently linked to baPWV $\geq$1600 cm/s. NAFLD patients were categorized into low-risk group (number of risk factors = 0), intermediate-risk group (= 1), and high-risk group ($\geq$2) based on their risk factors, including older age, hypertension, and biopsy-confirmed advanced fibrosis. The prevalence of baPWV $\geq$1600 cm/s was 7.1% (3/42) in the low-risk group, 30.8% (12/39) in the intermediate-risk group, and 63.9% (46/72) in the high-risk group. Non-invasive liver fibrosis markers and scores, including the FIB-4 index, NAFLD fibrosis score, hyaluronic acid, Wisteria floribunda agglutinin positive Mac-2-binding protein, and type IV collagen 7s, were feasible substitutes for invasive liver biopsy. Older age, hypertension, and advanced fibrosis are independently related to arterial stiffness, and a combination of these three factors may predict risk of arteriosclerosis in NAFLD patients.

**Funding:** The authors received no specific funding for this work.

**Competing interests:** The authors have declared that no competing interests exist.

## Introduction

Nonalcoholic fatty liver disease (NAFLD) is a major chronic liver disease, with a worldwide prevalence of approximately 25% [1, 2]. The disease is associated with the risk of progression to liver cirrhosis and hepatocellular carcinoma in some patients [3, 4]. The prognosis of NAFLD patients depends on the advancement of liver fibrosis [5–7] and is less favorable than that of healthy individuals. Furthermore, numerous patients die of cardiovascular disease (CVD) rather than liver-related events [6–8]. NAFLD is a multifactorial disease mutually associated with metabolic syndrome [9]. Reportedly, the presence of NAFLD is associated with subclinical atherosclerosis, independent from conventional metabolic risk factors [10–12]. However, the association between the advancement liver fibrosis and subclinical atherosclerosis remains controversial. In most of the previous studies, NAFLD was diagnosed based on abdominal ultrasonography and serum alanine aminotransferase (ALT) level, but not liver biopsy. Only a few studies have investigated the association between histological severity and subclinical atherosclerosis in biopsy-diagnosed NAFLD patients [13–15].

Recently, a genome-wide association study (GWAS) and subsequent related studies have demonstrated that single nucleotide polymorphisms (SNPs) in the patatin-like phospholipase domain containing 3 gene (*PNPLA3*) are associated with the development and severity of NAFLD [16–18]. The association between the *PNPLA3* SNP genotype and atherosclerosis in Italian NAFLD patients was previously reported [19]. Similarly, it has been confirmed in Japanese patients that the *PNPLA3* is a susceptibility gene involved in the development and advancement of NAFLD [20, 21], though its association with subclinical atherosclerosis has not been investigated.

In this study, arterial stiffness was evaluated using brachial-ankle pulse wave velocity (baPWV), and factors influencing arterial stiffness, including histological findings and the *PNPLA3* SNP, were investigated in Japanese biopsy-confirmed NAFLD patients. In addition, we performed a risk assessment for arteriosclerosis using clinical parameters.

## Materials and methods

### Patients

Among patients who visited Nippon Medical School Chiba Hokusoh Hospital and Nippon Medical School Hospital between August 2013 and July 2018, 153 patients aged 18 years or older underwent histological evaluation and were diagnosed with NAFLD, according to the European Association for the Study of the Liver guidelines as follows [22–24]: NAFLD was defined as the presence of steatosis in ≥5% of hepatocytes according to histological analysis. Exclusion criteria included 1) daily alcohol consumption ≥30 g for males and ≥20 g for females; 2) other chronic liver diseases, such as viral hepatitis B or C, autoimmune hepatitis, Wilson disease, and hemochromatosis; 3) secondary causes of steatosis, such as drug-induced fatty liver disease, total parenteral nutrition, and inborn errors of metabolism. A careful interview, clinical and laboratory evaluations, and image inspection were performed at the time of the liver biopsy in all patients.

The study protocol complied with the ethical guidelines established in accordance with the 2013 Declaration of Helsinki and was approved by the Ethics Committee of Nippon Medical School Chiba Hokusoh Hospital (approval number: 603). All patients provided written informed consent prior to entry into this study.

### Clinical and laboratory evaluation

Clinical and laboratory data were collected concurrently with liver biopsy. The body mass index (BMI) was calculated as weight (kg) divided by the square of height (m). Blood pressure

was measured in a seated position at least twice at an interval of several minutes, and the mean was calculated. Hypertension was diagnosed when systolic blood pressure was 135 mmHg or higher or diastolic blood pressure was 85 mmHg or higher and when patients were being treated with an antihypertensive drug [25]. Dyslipidemia was diagnosed when total cholesterol was 220 mg/dL or higher, high-density lipoprotein cholesterol (HDL cholesterol) was below 40 mg/dL, and/or triglycerides were 150 mg/dL or higher, as well as when patients were being treated with an antihyperlipidemic drug [26]. Type 2 diabetes was diagnosed according to the 2006 World Health Organization (WHO) criteria in addition to the presence of treatment with an oral hypoglycemic agent and insulin.

Laboratory evaluation included complete blood count, routine liver biochemistry (aspartate aminotransferase, ALT, total bilirubin, albumin, alkaline phosphatase, and gamma glutamyl transpeptidase), fasting lipids (total cholesterol, triglycerides, HDL cholesterol, and low-density lipoprotein cholesterol), fasting plasma glucose, hemoglobin A1c, and immunoreactive insulin. As an index of insulin resistance, the homeostasis model assessment-insulin resistance (HOMA-IR) was calculated using the following equation: HOMA-IR = fasting insulin (μU/mL) × plasma glucose (mg/dL)/405 [27]. Hyaluronic acid [28], type IV collagen 7s domain [29, 30], and Wisteria floribunda agglutinin positive Mac-2-binding protein (WFA$^+$-M2B) [30–32], all of which have been reported as useful liver fibrosis markers in NAFLD, were measured. In addition, the fibrosis scores such as the FIB-4 index [33] and NAFLD fibrosis scorer (NFS) [34] were calculated, as reported previously. DNA was extracted from each patient, and the *PNPLA3* rs738409 was genotyped by using a PCR protocol based on TaqMan assays.

## Pulse wave velocity

A noninvasive index of arterial stiffness, brachial-ankle PWV (baPWV), was measured using a volume-plethysmographic apparatus (form PWV/ABI; Colin, Co., Ltd., Komaki, Japan), under previously reported measurement conditions [35]: 1) the patients were examined after resting in the supine position for several minutes; 2) they refrained from ingesting caffeine and cigarette smoking starting 3 hours before measurement, as a rule; and 3) baPWV was measured in a quiet examination room controlled at a constant temperature (22°C–26°C). The mean of the bilateral baPWV values was used for analysis. baPWV was measured by skilled laboratory technicians who were blinded to patient information. Referring to previous reports, patients with baPWV ≥1,600 cm/s were defined as a risk group for cardiovascular events [36, 37].

## Histopathological evaluation

Histopathological evaluation was performed by experienced pathologists blinded to the clinical and laboratory data of the patients. NAFLD was diagnosed when lipid droplet deposition was noted in 5% or more hepatocytes. Then, steatosis, lobular inflammation, ballooning, and liver fibrosis were semi-quantitatively evaluated according to the NASH CRN scoring system [38]: steatosis was graded 0–3 based on the percent of hepatocytes on biopsy specimens (0: <5%, 1: 5–33%, 2: 33–66%, 3: >66%). Lobular inflammation was graded 0–3 based on inflammatory foci per 200× field (0: no foci, 1: <2 foci, 2: 2–4 foci, 3: >4 foci). Ballooning was graded 0–2 based on the number of hepatocytes with this change (0: none, 1: few cells, 2: many cells/prominent ballooning). Fibrosis stage was evaluated as follows: F0 = no fibrosis, F1 = perisinusoidal or periportal fibrosis, F2 = perisinusoidal and portal/periportal fibrosis, F3 = bridging fibrosis, and F4 = cirrhosis. F3–4 was provisionally designated as advanced fibrosis.

## Statistical analyses

Continuous variables were presented as medians and ranges, and categorical variables were presented as numbers and percentages. Continuous variables with skewed distribution were compared among or between groups using the Kruskal–Wallis test or the Mann–Whitney test, respectively. The Steel–Dwass test was applied when the Kruskal-Wallis test indicated a significant difference among groups. Multiple logistic regression analysis was used to identify the independent factors that were significantly associated with baPWV $\geq$1600 cm/s. The Cochran–Armitage test was used to investigate the changes in the prevalence of baPWV $\geq$1600 cm/s according to risk groups based on the number of risk factors including older age, hypertension, and advanced fibrosis. A receiver-operating characteristic (ROC) curve was generated in order to analyze the values of noninvasive markers and scores of fibrosis that most rationally predicted advanced fibrosis. All statistical analyses were performed using IBM SPSS version 17.0 (IBM Japan, Tokyo, Japan). The level of statistical significance was set at $p$ <0.05.

## Results

### Patients

Baseline characteristics of the 153 patients are shown in Table 1. There were 74 males and 79 females, and the median age was 57 years (range, 18–84 years). Regarding metabolic components, the median BMI was 28.8 kg/m$^2$ (range, 18.1–44.9 kg/m$^2$). There were 58 patients with type 2 diabetes (37.9%), 70 with hypertension (45.8%), and 117 with dyslipidemia (76.5%). On pathological examination of the 153 liver biopsy specimens, the fibrosis stage was determined to be F0 for 24 (15.7%) patients, F1 for 49 (32.0%), F2 for 36 (23.5%), F3 for 30 (19.6%), and F4 for 14 (9.2%) patients, with advanced fibrosis (F3–4) in 44 (28.8%) patients. The median baPWV value was 1557 cm/s (range, 1018–2776 cm/s). Among the 153 patients, 61 (39.9%) had a baPWV $\geq$1600 cm/s. The *PNPLA3* rs738409 was genotyped in 142 patients, and the distribution of the *PNPLA3* polymorphisms was as follows: CC, GC, and GG genotypes were found in 40.1% (57/142), 40.1% (57/142), and 19.7% (28/142) of patients, respectively.

### Relationship between baPWV and pathological severity of the disease

As shown in Fig 1, no correlation with baPWV was noted for inflammation and ballooning. On the other hand, baPWV significantly decreased with progression of liver steatosis ($p = 4.16 \times 10^{-2}$) and increased with progression of liver fibrosis ($p = 7.52 \times 10^{-3}$). When fibrosis stages were categorized into less advanced (F0-2) and advanced (F3-4) fibrosis groups, baPWV was significantly higher in the advanced fibrosis group ($p = 5.49 \times 10^{-4}$) (Fig 2).

### Factors associated with baPWV $\geq$1600 cm/s

Multiple logistic regression analysis showed that the following three variables were independently linked to baPWV $\geq$1600 cm/s (Table 2): older age ($\geq$55 years) ($p = 8.57 \times 10^{-3}$; OR = 3.03; 95% CI = 1.33–6.91), hypertension ($p = 1.05 \times 10^{-3}$; OR = 3.46; 95% CI = 1.65–7.28), and advanced fibrosis ($p = 9.22 \times 10^{-3}$; OR = 2.94; 95% CI = 1.31–6.63).

### Prevalence of baPWV $\geq$1600 cm/s according to the clinical risk scores based on older age, hypertension, and advanced fibrosis diagnosed by liver biopsy

We classified NAFLD patients into three groups: low-risk group (number of risk factors = 0), intermediate-risk group (= 1), and high-risk group ($\geq$2) based on the number of risk factors

**Table 1. Baseline characteristics of the 153 patients.**

| Factors | n = 153 |
|---|---|
| Age (year) | 57 (18–84) |
| Gender (M/F) | 74/79 |
| BMI (kg/m$^2$) | 28.8 (18.1–44.9) |
| Platelets (×10$^3$/mm$^3$) | 199 (51–411) |
| AST (U/L) | 54 (19–183) |
| ALT (U/L) | 70 (10–401) |
| γ-GTP (U/L) | 63 (15–488) |
| Serum albumin (g/dL) | 4.0 (2.5–5.2) |
| Prothrombin time (%) | 95.4 (41.7–138.5) |
| Total cholesterol (mg/dL) | 193 (96–312) |
| HDL cholesterol (mg/dL) | 47 (23–96) |
| Triglyceride (mg/dL) | 138 (42–480) |
| Plasma glucose (mg/dL) | 105 (78–333) |
| Insulin (μU/mL) | 11.9 (1.7–88.5) |
| HOMA-IR | 3.50 (0.57–40.6) |
| Type 2 diabetes (presence/absence) | 58/95 |
| Hypertension (presence/absence) | 70/83 |
| Smoking (yes/no) | 63/90 |
| baPWV (cm/s) | 1557 (1018–2776) |
| Hyaluronic acid (ng/ml) | 50.0 (10.0–3284) |
| Type IV collagen 7s (ng/ml) | 4.7 (2.5–12.7) |
| WFA$^+$-M2BP (C.O.I) | 0.9 (0.24–8.3) |
| FIB-4 index | 2.17 (0.32–10.5) |
| NFS | -0.61 (-4.52–3.80) |
| *PNPLA3* genotype (CC/GC/GG/unknown) | 57/57/28/11 |
| Liver steatosis (1/2/3) | 81/57/15 |
| Liver inflammation (0/1/2/3) | 7/89/52/5 |
| Liver ballooning (0/1/2) | 31/101/21 |
| Liver fibrosis stage (F0/F1/F2/F3/F4) | 24/49/36/30/14 |

Data are presented as numbers or median (range).

BMI, body mass index; AST, aspartate aminotransferase; ALT, alanine aminotransferase; γ-GTP, gamma glutamyl transpeptidase; HDL, high-density lipoprotein; HOMA-IR, homeostasis model assessment-insulin resistance; baPWV, brachial-ankle pulse wave velocity; WFA$^+$-M2BP, Wisteria floribunda agglutinin positive Mac-2-binding protein; FIB-4, fibrosis-4; NFS, NAFLD (nonalcoholic fatty liver disease) fibrosis score; *PNPLA3*, patatin-like phospholipase domain containing 3.

linked independently to baPWV ≥1600 cm/s, including older age, hypertension, and advanced fibrosis, as described above. The prevalence of baPWV ≥1600 cm/s was 7.1% (3/42) in the low-risk group, 30.8% (12/39) in the intermediate-risk group, and 63.9% (46/72) in the high-risk group, respectively ($p = 9.44 \times 10^{-10}$) (Fig 3).

## Prevalence of baPWV ≥1600 cm/s according to the clinical risk scores based on older age, hypertension, and advanced fibrosis as diagnosed by fibrosis markers and scores

Using ROC analyses for the diagnosis of advanced fibrosis, the area under the curve (AUC) of the fibrosis markers and scores were as follows: FIB-4 index (cut-off value = 2.49,

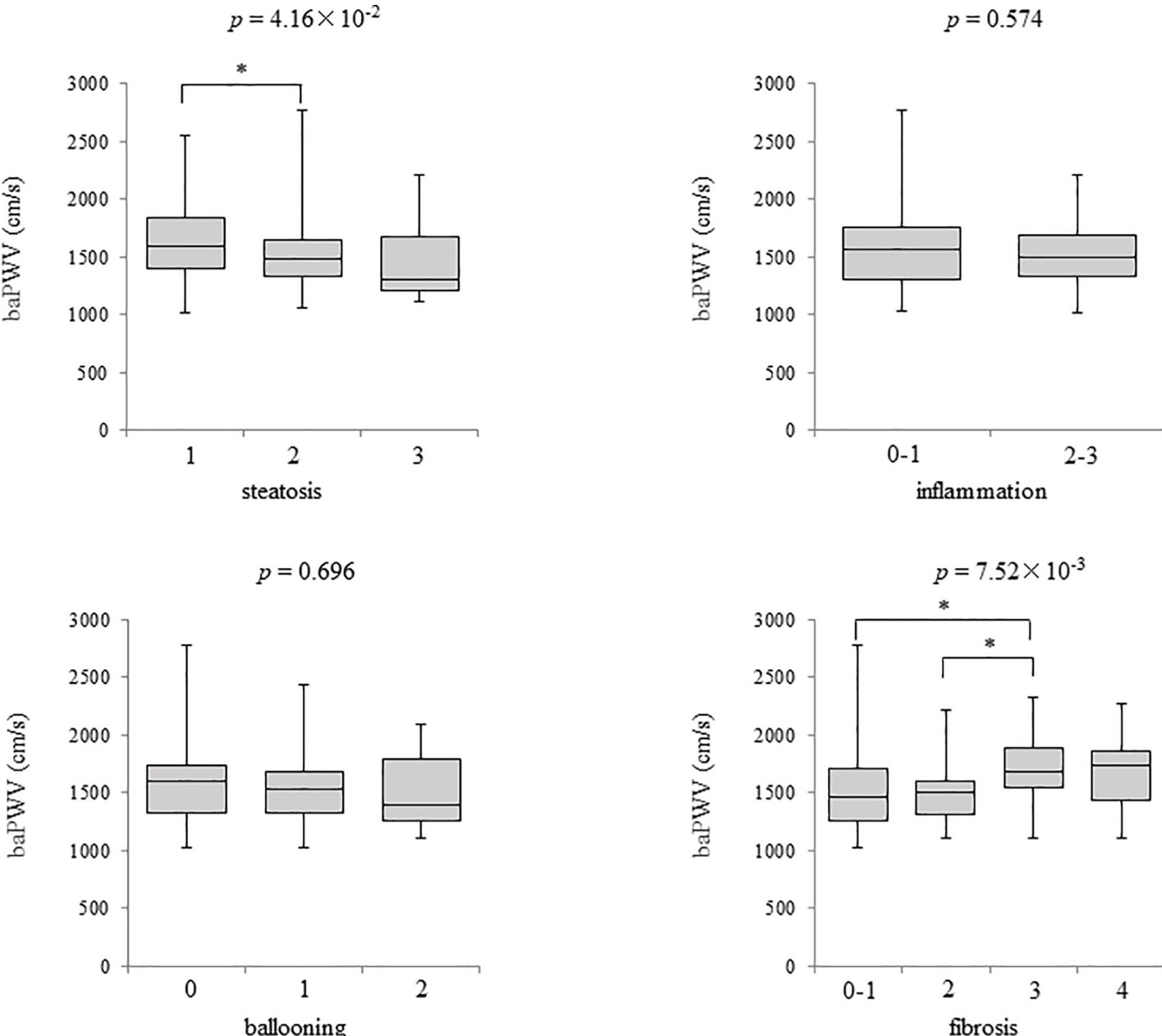

**Fig 1. Box and whisker plots of baPWV values according to the severity of each histological component in NAFLD patients.** baPWV, brachial-ankle pulse wave velocity; NAFLD, nonalcoholic fatty liver disease. *, p < 0.05.

AUC = 0.836), NFS ($9.47 \times 10^{-2}$, 0.874), hyaluronic acid (62.3 ng/mL, 0.871), WFA$^+$-M2BP (0.95 C.O.I, 0.816), and type IV collagen 7s (5.2 ng/mL, 0.842) (Fig 4). Next, we reclassified NAFLD patients into three risk groups based on the number of risk factors, including older age, hypertension, and advanced fibrosis, as diagnosed by each noninvasive fibrosis marker and score instead of by invasive liver biopsy. The prevalence of NAFLD patients with baPWV $\geq$1600 cm/s increased with a greater number of risk factors, even when advanced fibrosis was diagnosed by FIB-4 index ($p = 9.25 \times 10^{-6}$), NFS ($p = 7.01 \times 10^{-8}$), hyaluronic acid ($p = 3.31 \times 10^{-7}$), WFA$^+$-M2BP ($p = 2.49 \times 10^{-7}$), and type IV collagen 7s ($p = 7.30 \times 10^{-7}$).

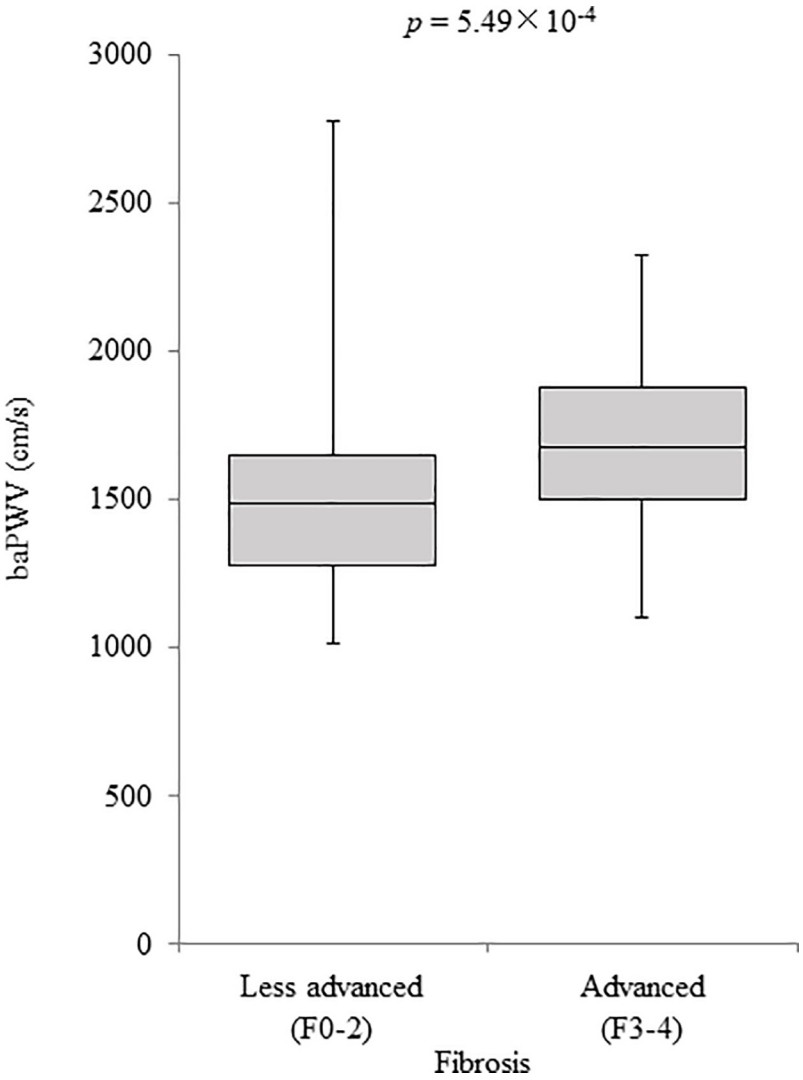

$p = 5.49 \times 10^{-4}$

**Fig 2. baPWV values in NAFLD patients according to fibrosis stage.** baPWV values (median, 1489 cm/s) in the advanced fibrosis group (fibrosis stage = F3-4) were significantly higher than those (median, 1679 cm/s) in the less advanced fibrosis group (fibrosis stage = F0-2) ($p = 5.49 \times 10^{-4}$). baPWV, brachial-ankle pulse wave velocity; NAFLD, nonalcoholic fatty liver disease.

## Discussion

In this study, we clarified that older age, hypertension, and advanced liver fibrosis were independently associated with arterial stiffness in Japanese biopsy-proven NAFLD patients. Arterial stiffness is evaluated by measuring PWV, which is widely used as a preclinical cardiovascular risk marker [39]. Several studies reported that the PWV in NAFLD patients is higher than that in healthy individuals and that the presence of NAFLD is associated with arterial stiffness independent from conventional metabolic risk factors [40–44]. On the other hand, only a few studies have investigated the association of the severity of NAFLD with arterial stiffness. Advanced fibrosis estimated based on NFS [45] and transient elastography [46] were reported to be associated with a high PWV value independent from conventional metabolic risk factors in patients diagnosed with NAFLD by ultrasonography. However, to our knowledge, the association between the severity of liver disease and PWV was histologically

**Table 2. Univariate and multivariate logistic regression analysis of factors associated with baPWV ≥1600 cm/s.**

| Factors | Category | Univariate | | | Multivariate | | |
|---|---|---|---|---|---|---|---|
| | | OR | 95% CI | *p* value | OR | 95% CI | *p* value |
| Age (years) | Older age (≥55) | 4.86 | 2.29–10.32 | $3.84 \times 10^{-5}$ | 3.03 | 1.33–6.91 | $8.57 \times 10^{-3}$ |
| Gender | Female | 1.32 | 0.69–2.52 | 0.683 | | | |
| BMI (kg/m²) | By 1 kg/m² down | 1.04 | 0.97–1.12 | 0.274 | | | |
| Total-cholesterol (mg/dL) | By 1 mg/dL down | 1.01 | 1.00–1.02 | 0.194 | | | |
| HDL-cholesterol (mg/dL) | By 1 mg/dL up | 1.01 | 0.98–1.04 | 0.621 | | | |
| Triglyceride (mg/dL) | By 1 mg/dL down | 1.00 | 1.00–1.01 | 0.129 | | | |
| Plasma glucose (mg/dL) | By 1 mg/dL up | 1.00 | 0.99–1.01 | 0.716 | | | |
| Insulin (μU/mL) | By 1 μU/mL up | 1.02 | 0.98–1.05- | 0.345 | | | |
| HOMA-IR | By 1 up | 1.04 | 0.95–1.13 | 0.402 | | | |
| Diabetes | Presence | 1.56 | 0.80–3.04 | 0.188 | | | |
| Hypertension | Presence | 4.45 | 2.23–8.90 | $2.35 \times 10^{-5}$ | 3.46 | 1.65–7.28 | $1.05 \times 10^{-3}$ |
| Smoking | no | 1.13 | 0.59–2.19 | 0.708 | | | |
| *PNPLA3* genotype | GG | 1.53 | 0.67–3.52 | 0.313 | | | |
| Liver steatosis | 1 grade down | 1.53 | 0.92–2.56 | $9.92 \times 10^{-2}$ | | | |
| Liver inflammation | 1 grade down | 1.23 | 0.73–2.08 | 0.438 | | | |
| Liver ballooning | 1 grade down | 1.18 | 0.67–2.06 | 0.567 | | | |
| Liver fibrosis stage | Advanced fibrosis | 4.65 | 2.20–9.82 | $5.53 \times 10^{-5}$ | 2.94 | 1.31–6.63 | $9.22 \times 10^{-3}$ |

baPWV, brachial-ankle pulse wave velocity; OR, odds ratio; CI, confidence interval; BMI, body mass index, HDL, high-density lipoprotein; HOMA-IR, homeostasis model assessment-insulin resistance; *PNPLA3*, patatin-like phospholipase domain containing 3.

investigated by liver biopsy in only 2 reports of a small number of patients from Turkey, with contradictory results; while the progression of histological liver fibrosis was an independent factor for a high PWV value in a study involving 100 biopsy-confirmed NAFLD patients [13], no difference was noted in PWV between patients with simple steatosis and patients with steatohepatitis in the other study involving 61 NAFLD patients [14]. Our analysis of a relatively large Japanese cohort was comparable to and supported the findings of the former study. In this study, baPWV was elevated with the progress of liver fibrosis, while it was decreased with the progress of hepatic steatosis. This paradox can be explained by the loss of hepatic fat in NAFLD patients with advanced fibrosis. This result may suggest that liver fibrosis affects the arterial stiffness more than the hepatic fat.

The arterial stiffness-promoting mechanism of the presence and severity of NAFLD independent of conventional metabolic risk factors remains unclear, though there is experimental evidence supporting that NAFLD and arterial stiffness develop and progress due to a common etiology [47–49]. First, chronic inflammation and oxidative stress, considered important factors for the development and progression of NAFLD, induce cardiovascular disorder. Second, it has been reported that the blood level of adiponectin, which has anti-inflammatory and anti-fibrosis activity, decreases due to an increase in adipose tissue and chronic inflammation, and this decrease then promotes NAFLD and arterial stiffness. Third, an influence of TGF-β, which plays an important role in the progression of liver fibrosis, on arterial stiffness has been suggested. To clarify the association between NAFLD and arterial stiffness, further studies are necessary.

To the best of our knowledge, this is the first report to analyze factors associated with arterial stiffness, including the *PNPLA3* SNP genotype. Two studies on the association between the carotid intima-media thickness, a surrogate marker of subclinical atherosclerosis as is PWV, and the *PNPLA3* SNP genotype in Italian NAFLD patients have been reported, but the

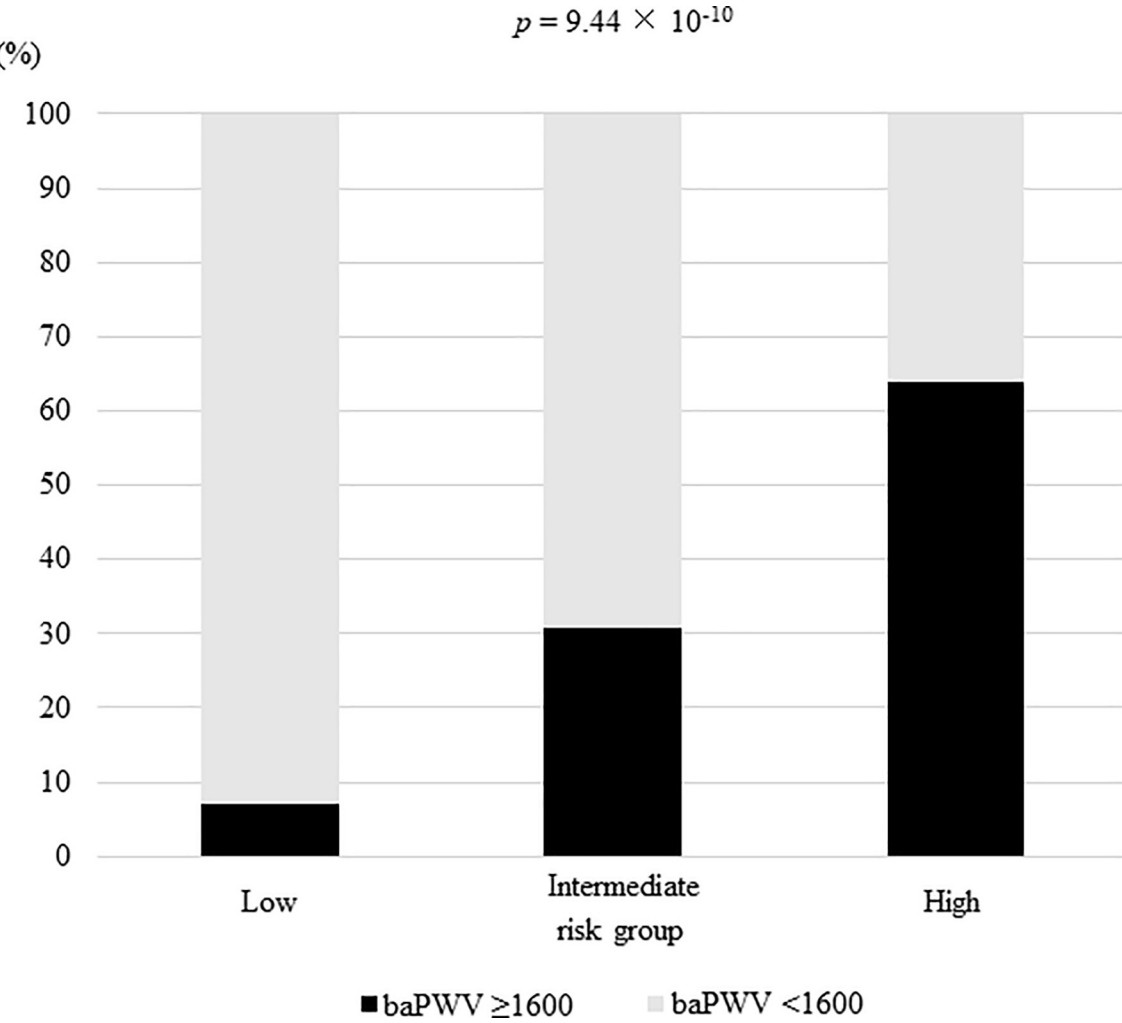

**Fig 3. The prevalence of baPWV ≥1600 cm/s according to risk groups based on the number of risk factors including older age, hypertension, and advanced fibrosis; low-risk group (number of risk factors = 0), intermediate-risk group (= 1), and high-risk group (≥2).** baPWV, brachial-ankle pulse wave velocity.

findings were contradictory. Petta et al. reported that the *PNPLA3* GG genotype was significantly associated with the severity of carotid atherosclerosis in young NAFLD patients [19], but Di Costanzo et al. found no association between the carotid intima-media thickness and the *PNPLA3* SNP genotype and indicated that complications due to metabolic abnormalities influenced the carotid intima-media thickness, which is well known [50]. In our study, no association was noted between baPWV and the *PNPLA3* SNP genotype, but further investigation is necessary regarding the influence of the *PNPLA3* SNP genotype on atherosclerosis in light of racial differences in the morbidity of atherosclerosis and the *PNPLA3* SNP genotype distribution.

In this study, older age, hypertension, and advanced fibrosis were additively related to arterial stiffness in NAFLD patients, and it is possible to speculate the risk of arteriosclerosis progression by combining these three factors. Furthermore, we showed that the risk assessment of atherosclerosis progression in clinical practice is possible by substituting noninvasive liver fibrosis markers and scores for histological diagnosis by invasive liver biopsy. Clinicians

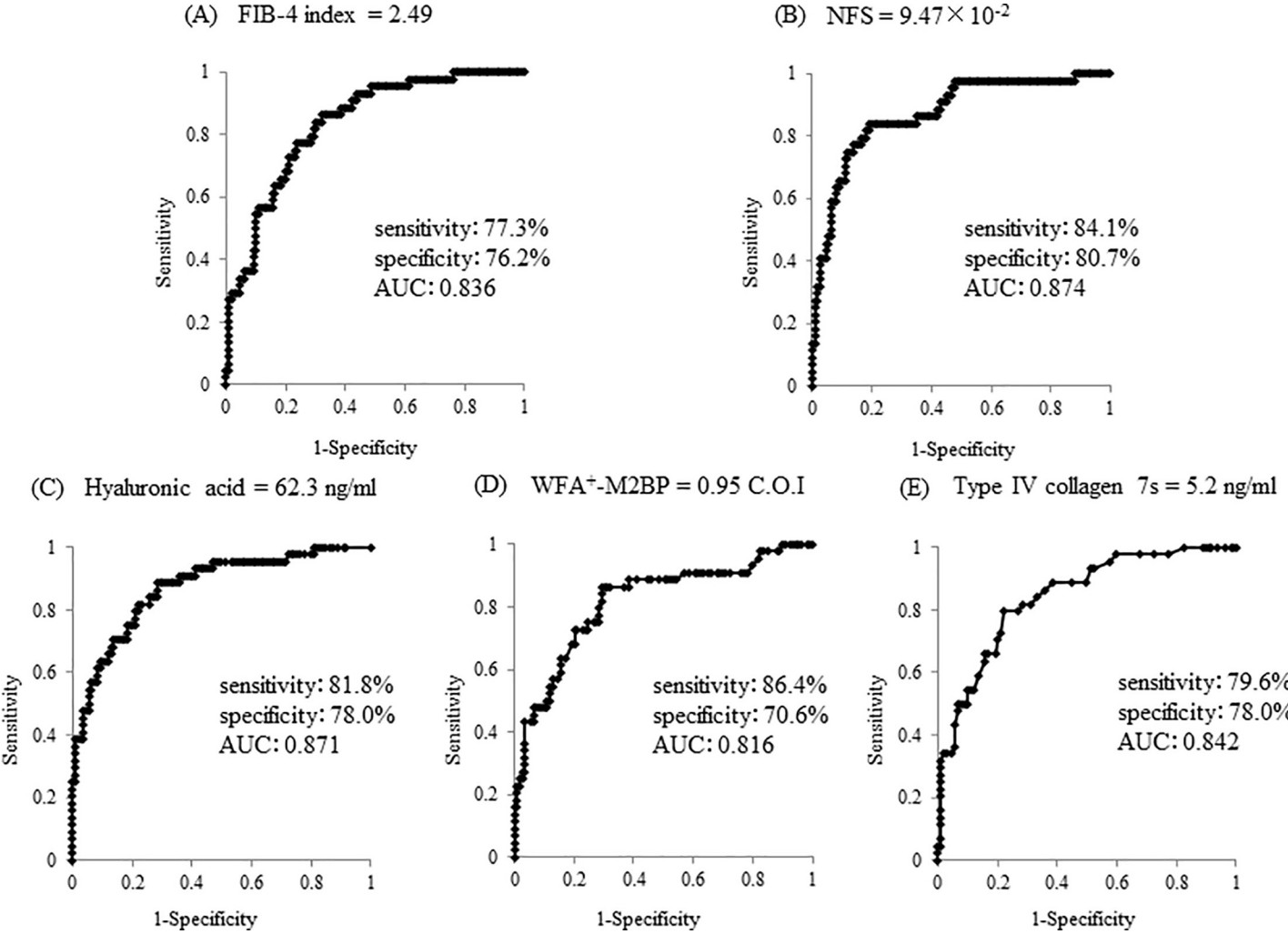

**Fig 4.** Receiver-operating characteristic (ROC) curves of FIB4-index (A), NFS (B), hyaluronic acid (C), WFA+-M2BP (D), and type IV collagen 7s (E) for predicting advanced fibrosis (F3-4). FIB-4, fibrosis-4; NFS, NAFLD (nonalcoholic fatty liver disease) fibrosis score; WFA+-M2BP, Wisteria floribunda agglutinin positive Mac-2-binding protein.

should pay attention to cardiovascular events in NAFLD patients with high fibrosis marker levels and scores as well as older age and hypertension, and further care should be taken when these factors overlap.

There were some limitations in this study. First, the number of patients, especially those with advanced fibrosis, was relatively small. Second, as described above, there are racial differences in atherosclerosis and the PNPLA3 genotype. To make a definitive conclusion, it may be necessary to re-confirm the results of this study in independent validation cohorts with different characteristics, races, and/or ethnicities.

In conclusion, older age, hypertension, and advanced liver fibrosis were found to be independent factors associated with arterial stiffness in Japanese biopsy-proven NAFLD patients. Furthermore, the combination of older age, hypertension, and advanced fibrosis based on noninvasive fibrosis markers and scores may predict the risk of arteriosclerosis progression in NAFLD patients in clinical practice.

## Supporting information

**S1 Fig.** Box and whisker plots of baPWV values according to each fibrosis marker and score such as FIB-4 index (A), NFS (B), hyaluronic acid (C), WFA+-M2BP (D), and type IV collagen 7s (E). baPWV, brachial-ankle pulse wave velocity; FIB-4, fibrosis-4; NFS, NAFLD (nonalcoholic fatty liver disease) fibrosis score; WFA+-M2BP, Wisteria floribunda agglutinin positive Mac-2-binding protein.
(TIF)

**S1 File. Exel.** Raw data including pathological findings and baPWV.
(XLSX)

## Acknowledgments

The authors wish to thank all medical doctors from all institutions who were involved in this study.

## Author Contributions

**Data curation:** Tadamichi Kawano, Mai Koeda, Tomohide Tanabe, Tomomi Okubo, Korenobu Hayama, Ai Iwashita, Norio Itokawa, Keiko Kaneko, Tsutomu Hatori, Naoya Emoto, Etsuko Iio, Yasuhito Tanaka.

**Formal analysis:** Taeang Arai, Chisa Kondo.

**Investigation:** Taeang Arai, Tadamichi Kawano, Mai Koeda, Yuji Yoshida, Tomomi Okubo, Korenobu Hayama, Ai Iwashita, Norio Itokawa, Chisa Kondo, Keiko Kaneko, Tsutomu Hatori, Etsuko Iio.

**Supervision:** Akihito Tsubota, Chiaki Kawamoto, Yasuhito Tanaka, Katsuhiko Iwakiri.

**Writing – original draft:** Taeang Arai, Masanori Atsukawa.

**Writing – review & editing:** Masanori Atsukawa, Akihito Tsubota.

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
