## [Decision Letter · Decision Letter 0]

21 Aug 2019

PONE-D-19-17091

Factors influencing subclinical atherosclerosis in patients with biopsy-proven nonalcoholic fatty liver disease

PLOS ONE

Dear Dr. Atsukawa,

Thank you for submitting your manuscript to PLOS ONE. After careful consideration, we feel that it has merit but does not fully meet PLOS ONE’s publication criteria as it currently stands. Therefore, we invite you to submit a revised version of the manuscript that addresses the points raised during the review process.

We would appreciate receiving your revised manuscript by Oct 05 2019 11:59PM. To enhance the reproducibility of your results, we recommend that if applicable you deposit your laboratory protocols in protocols.io, where a protocol can be assigned its own identifier (DOI) such that it can be cited independently in the future. For instructions see: http://journals.plos.org/plosone/s/submission-guidelines#loc-laboratory-protocols

We look forward to receiving your revised manuscript.

Kind regards,

Jee-Fu Huang, M.D., Ph.D.

Academic Editor

PLOS ONE

Journal Requirements:

1. Please amend your authorship list in your manuscript file to include author  Norio Iwashita

2. Please amend the manuscript submission data (via Edit Submission) to include author Norio Itokawa

Reviewers' comments:

Reviewer's Responses to Questions

5. Review Comments to the Author

Reviewer #1: Arai T et al evaluated the factors influencing atherosclerosis in patients with biopsy-proven nonalcoholic fatty liver disease (NAFLD), and found advavalunced liver fibrosis was associated with arterial stiffness. The study is interesting, however, some issues need to be addressed.

Abstract:

1. Line 42-43: Please show the value of 95% CI.

2. Line 48: Please show the value of p.

Methods:

1. Line 119-123: Hyaluronic acid [28], type IV 120 collagen 7s domain [29, 30], and Wisteria floribunda agglutinin positive Mac-2-binding protein (WFA+-M2B) [30–32], all of which have been reported as useful liver fibrosis markers in NAFLD, were measured. In addition, the fibrosis scores such as the FIB-4 index [33] and NAFLD fibrosis scorer (NFS) [34] were calculated, as reported previously. please change to Hyaluronic acid [28], type IV collagen 7s domain [29, 30], Wisteria floribunda agglutinin positive Mac-2-binding protein (WFA+-M2B) [30–32], FIB-4 index [33] and NAFLD fibrosis scorer (NFS) were measured. It is not suitable to write comments in the Methods.

Results:

1. How about NFS score in analysis, including Table 1, Table 2 and Figure 1?

2. In this study, baPWV was used without considering ABI values. It has been widely recognized that baPWV value is meaning less to measure in the limb of ABI value <0.9. How are the results if you exclude baPWV data associated with ABI <0.9? Addition, how about the association between ABI and liver fibrosis stage?

3. Line 280-281: This paradox can be explained by the loss of hepatic fat in NAFLD patients with advanced fibrosis. This result may suggest that liver fibrosis affects the arterial stiffness more than the hepatic fat. This explanation is too simple to convince others.

4. Line 323: based on noninvasive fibrosis markers and scores may predict the risk of arteriosclerosis� please change to arterial stiffness. After all, the two nouns are different.

Reviewer #2: This is a well conducted study examining the association of the severity of NAFLD and subclinical atherosclerosis in 153 biops-proven NAFLD patients. The findings are interesting and important, while some points need to be further clarified.

1. Patients who underwent histological evaluation were eligible and included in this study. However, the reasons for histological evaluation are unclear. The authors need to show the reasons for histological examination of the study subjects.

2. This study shows an association between liver fibrosis and subclinical atherosclerosis, but not a causal association. Therefore, the statement "...liver fibrosis affects the arterial stiffness...." is incorrect.

3. The results in paragraphs "Prevalence of baPWV ≥1600 cm/s according to the clinical risk scores based on older age, hypertension, and advanced fibrosis diagnosed by liver biopsy" and "Prevalence of baPWV ≥1600 cm/s according to the clinical risk scores based on older age, hypertension, and advanced fibrosis as diagnosed by fibrosis markers and scores" are redudant and may be removed. Moreover, the authors need to exaplain why they chose a cut-off level of baPWV as 1600 cm/s.

4. What kinds of drugs that may induce fatty liver were examined and excluded in this study?

5. The authors need to exaplain why this study could not show the associations of metabolic factors/ and related disorders (sucah as daibetes mellitus and hyperlipidemia...) with liver fibrosis.

Reviewer #3: PONE-D-19-17091

Taeang Arai et al., Factors influencing subclinical atherosclerosis in patients with biopsy-proven nonalcoholic fatty liver disease

This study aimed to clarify the factors related to subclinical arteriosclerosis, including the histopathological severity of the disease and PNPLA3 gene 38 polymorphisms, in 153 biopsy-proven nonalcoholic fatty liver disease (NAFLD) patients. The authors found that older age (≥55 years), hypertension, and advanced fibrosis were independently linked to baPWV ≥1600 cm/s. The study is appropriately designed and executed. However, there are some concerns to be clarified:

1. The association between subclinical atherosclerosis and NAFLD, indeed, has been studied extensively in the past. Retrospective and prospective studies have provided evidences of a strong association between NAFLD and subclinical atherosclerosis, including increased intima-media thickness, endothelial dysfunction, arterial stiffness, impaired left ventricular function and coronary calcification. (Fargion S et al., World J Gastroenterol 2014;20:13306-24.). The relevant meta-analysis has also been published (Ampuero J et al., Rev Esp Enferm Dig;107:10-6.). A recent study has also addressed this issue (Gill C et al., Am J Cardiol 2017;119:1717-22.).

2. Also, the progression of histological liver fibrosis as an independent factor for a high PWV value in a study involving 100 biopsy-confirmed 276 NAFLD patients has been reported [Ref. 13 of the manuscript]. The authors also mentioned that the only difference of their study as compared with the previous studies is that theirs is a “relatively large Japanese cohort” (Line 1, Page 17, the manuscript), which made this study not novel enough.

3. Figure 1 shows that no correlation with baPWV was noted for histological findings of liver inflammation and ballooning. Multiple logistic regression analysis showed that the following the only histological variable independently linked to baPWV ≥1600 cm/s (Table 2) was advanced fibrosis. This raises the concern if baPWV is actually related only to liver fibrosis, but not necessary to steatosis (inflammation and ballooning). The previous study has indicated that the carotid intima thickness, a parameter of subclinical atherosclerosis, significantly increased in chronic hepatitis C virus patients especially in those with cirrhosis and closely correlated with each other (Barakat AAE et al., Egypt Heart J 2017;69:139-47.).

4. As discussed in the Paragraph 2, Discussion of the manuscript, the arterial stiffness-promoting mechanism of the presence and severity of NAFLD independent of conventional metabolic risk factors remains unclear. To strengthen the novelty of this study, the levels of parameters of chronic inflammation, oxidative stress, adiponectin and TGF-β can be checked.

5. Although single nucleotide polymorphisms (SNPs) in the patatin-like phospholipase domain containing 3 gene (PNPLA3) are associated with the development and severity of NAFLD [Ref. 16–18 of the manuscript], this study only mentioned the distribution of the PNPLA3 polymorphisms without showing the association among PNPLA3 polymorphisms and severity of NAFLD.

6. It would be interesting to have sub-group analyses to search for any further possible correlations. For instance, although the current study did not identify an association between baPWV and the PNPLA3 SNP genotype, sub-group analyses according to gender or age may exert different findings.

---

## [Author Response · Author response to Decision Letter 0]

12 Sep 2019

Reviewer #1: Arai T et al evaluated the factors influencing atherosclerosis in patients with biopsy-proven nonalcoholic fatty liver disease (NAFLD), and found advanced liver fibrosis was associated with arterial stiffness. The study is interesting, however, some issues need to be addressed.

→We wish to express our strong appreciation to the reviewer for insightful comments on our paper. We feel the comments have helped us significantly improve the paper.

Abstract:

1. Line 42-43: Please show the value of 95% CI.

→As suggested by the reviewer, we added the value of 95% CI in line 42-44.

2. Line 48: Please show the value of p.

→As suggested by the reviewer, we added the value of p in line 49.

Methods:

1. Line 119-123: Hyaluronic acid [28], type IV 120 collagen 7s domain [29, 30], and Wisteria floribunda agglutinin positive Mac-2-binding protein (WFA+-M2B) [30–32], all of which have been reported as useful liver fibrosis markers in NAFLD, were measured. In addition, the fibrosis scores such as the FIB-4 index [33] and NAFLD fibrosis scorer (NFS) [34] were calculated, as reported previously. please change to Hyaluronic acid [28], type IV collagen 7s domain [29, 30], Wisteria floribunda agglutinin positive Mac-2-binding protein (WFA+-M2B) [30–32], FIB-4 index [33] and NAFLD fibrosis scorer (NFS) were measured. It is not suitable to write comments in the Methods.

→The text has been revised as instructed by the reviewer.

Results:

1. How about NFS score in analysis, including Table 1, Table 2 and Figure 1?

→Table 1 already included the median and range of NFS. When reanalyzing in Table2, including NFS, NFS was extracted as a significant factor related to baPWV≥1600 cm/s in univariate analysis (p = 1.01×10-3; OR = 1.40 ;95% CI = 1.15-1.71), but not as a significant factor in multivariate analysis (p = 0.55; OR = 0.91 ;95% CI = 0.66-1.24). As suggested by the reviewer, we analyzed the relationship between baPWV and each fibrosis marker and score including NFS and presented in supplementary figure 1 (line 241-243).

2. In this study, baPWV was used without considering ABI values. It has been widely recognized that baPWV value is meaning less to measure in the limb of ABI value <0.9. How are the results if you exclude baPWV data associated with ABI <0.9? Addition, how about the association between ABI and liver fibrosis stage?

→Of the 153 patients, there was only one patient with ABI value <0.9. Therefore, in this study, we might not need to consider ABI. There was also a report on the relationship between the presence of NAFLD and ABI [J Diabetes. 2017 Jun;9(6):586-595.], and as you pointed out, we investigated the association between ABI and liver fibrosis stage. However, in this study, there was no significant correlation between ABI and liver fibrosis stage.

3. Line 280-281: This paradox can be explained by the loss of hepatic fat in NAFLD patients with advanced fibrosis. This result may suggest that liver fibrosis affects the arterial stiffness more than the hepatic fat. This explanation is too simple to convince others.

→As the reviewer pointed out, we have made a major correction in this part (line 280-299).

4. Line 323: based on noninvasive fibrosis markers and scores may predict the risk of arteriosclerosis please change to arterial stiffness. After all, the two nouns are different.

→As suggested by the reviewer, we rephrased " arteriosclerosis " to " arterial stiffness "(line 348)

Reviewer #2: This is a well conducted study examining the association of the severity of NAFLD and subclinical atherosclerosis in 153 biops-proven NAFLD patients. The findings are interesting and important, while some points need to be further clarified.

→We are deeply grateful to the reviewer for the critical comments and applaud the careful reading of our manuscript.

1. Patients who underwent histological evaluation were eligible and included in this study. However, the reasons for histological evaluation are unclear. The authors need to show the reasons for histological examination of the study subjects.

→As the reviewer pointed out, the reason for the liver biopsy was specified in line 139-141.

2. This study shows an association between liver fibrosis and subclinical atherosclerosis, but not a causal association. Therefore, the statement "...liver fibrosis affects the arterial stiffness...." is incorrect.

→As the reviewer pointed out, we have revised the text.

3. The results in paragraphs "Prevalence of baPWV ≥1600 cm/s according to the clinical risk scores based on older age, hypertension, and advanced fibrosis diagnosed by liver biopsy" and "Prevalence of baPWV ≥1600 cm/s according to the clinical risk scores based on older age, hypertension, and advanced fibrosis as diagnosed by fibrosis markers and scores" are redudant and may be removed. Moreover, the authors need to exaplain why they chose a cut-off level of baPWV as 1600 cm/s.

→As the reviewer pointed out, the result part is redundant, so figure 5 has been deleted. We mentioned that patients with baPWV ≥1,600 cm/s were defined as a risk group for cardiovascular events according to previous reports in line 135-136, but we have now also added this to the results section for clarity (line 212).

4. What kinds of drugs that may induce fatty liver were examined and excluded in this study?

→Patients taking amiodarone and tamoxifen which may induce fatty liver were excluded from the study (Line 94). 

5. The authors need to explain why this study could not show the associations of metabolic factors/ and related disorders (sucah as daibetes mellitus and hyperlipidemia...) with liver fibrosis.

→As the reviewer pointed out, metabolic factors/ and related disorders are important factors related to liver fibrosis in NAFLD patients, but this study does not analyze factors related to liver fibrosis.

Reviewer #3: PONE-D-19-17091

Taeang Arai et al., Factors influencing subclinical atherosclerosis in patients with biopsy-proven nonalcoholic fatty liver disease

This study aimed to clarify the factors related to subclinical arteriosclerosis, including the histopathological severity of the disease and PNPLA3 gene 38 polymorphisms, in 153 biopsy-proven nonalcoholic fatty liver disease (NAFLD) patients. The authors found that older age (≥55 years), hypertension, and advanced fibrosis were independently linked to baPWV ≥1600 cm/s. The study is appropriately designed and executed. However, there are some concerns to be clarified:

→We thank you for your comments, which have helped us to greatly improve the manuscript

1. The association between subclinical atherosclerosis and NAFLD, indeed, has been studied extensively in the past. Retrospective and prospective studies have provided evidences of a strong association between NAFLD and subclinical atherosclerosis, including increased intima-media thickness, endothelial dysfunction, arterial stiffness, impaired left ventricular function and coronary calcification. (Fargion S et al., World J Gastroenterol 2014;20:13306-24.). The relevant meta-analysis has also been published (Ampuero J et al., Rev Esp Enferm Dig;107:10-6.). A recent study has also addressed this issue (Gill C et al., Am J Cardiol 2017;119:1717-22.).

2. Also, the progression of histological liver fibrosis as an independent factor for a high PWV value in a study involving 100 biopsy-confirmed 276 NAFLD patients has been reported [Ref. 13 of the manuscript]. The authors also mentioned that the only difference of their study as compared with the previous studies is that theirs is a “relatively large Japanese cohort” (Line 1, Page 17, the manuscript), which made this study not novel enough.

1,2

→As pointed out by the reviewer, there have been numerous reports on the relation between the presence of NAFLD and subclinical arteriosclerosis. However, only a few studies have showed the association between histological severity and subclinical atherosclerosis in biopsy-diagnosed NAFLD patients. Unlike other reports, we analyzed the relation between arterial stiffness and severity of NAFLD including not only liver fibrosis but also steatosis, lobular inflammation, ballooning and our results suggested that liver fibrosis was the only possible histological feature associated with arterial stiffness. In addition, for the first time, we also found that the age and hypertension well known to be associated with arterial stiffness and advanced fibrosis are additively related to PWV, and it is possible to speculate the risk of arteriosclerosis progression by combining these three factors.

3. Figure 1 shows that no correlation with baPWV was noted for histological findings of liver inflammation and ballooning. Multiple logistic regression analysis showed that the following the only histological variable independently linked to baPWV ≥1600 cm/s (Table 2) was advanced fibrosis. This raises the concern if baPWV is actually related only to liver fibrosis, but not necessary to steatosis (inflammation and ballooning). The previous study has indicated that the carotid intima thickness, a parameter of subclinical atherosclerosis, significantly increased in chronic hepatitis C virus patients especially in those with cirrhosis and closely correlated with each other (Barakat AAE et al., Egypt Heart J 2017;69:139-47.).

→As the reviewer suggested, we re-considered the result that only liver fibrosis in the histological features was associated with arterial stiffness (line 280-299).

4. As discussed in the Paragraph 2, Discussion of the manuscript, the arterial stiffness-promoting mechanism of the presence and severity of NAFLD independent of conventional metabolic risk factors remains unclear. To strengthen the novelty of this study, the levels of parameters of chronic inflammation, oxidative stress, adiponectin and TGF-β can be checked.

→Although reviewer's proposal is very valuable for clarifying the relationship between the presence and severity of NAFLD and arterial stiffness, unfortunately, the levels of parameters of chronic inflammation, oxidative stress, adiponectin and TGF-β could not be measured in this study. We added these limitations to Discussion section (line 312-314).

5. Although single nucleotide polymorphisms (SNPs) in the patatin-like phospholipase domain containing 3 gene (PNPLA3) are associated with the development and severity of NAFLD [Ref. 16–18 of the manuscript], this study only mentioned the distribution of the PNPLA3 polymorphisms without showing the association among PNPLA3 polymorphisms and severity of NAFLD.

→Thank you for a very important suggestion. We performed additional analysis of the association between the histological severity of NAFLD including histological findings and PNPLA3 genotype, but there was no significant association in this study. As one of the reasons, we speculate that the number of patients with the PNPLA3 GG genotype, which is a risk genotype of the development and severity of NAFLD, was relatively small in this study. We added these limitations in Discussion section (line 341-343).

6. It would be interesting to have sub-group analyses to search for any further possible correlations. For instance, although the current study did not identify an association between baPWV and the PNPLA3 SNP genotype, sub-group analyses according to gender or age may exert different findings.

→As suggested by the reviewer, various sub-group analyses including gender and age were performed, but no significant correlation was found between PNPLA3 genotype and baPWV. As mentioned above, we think that small number of patients with the PNPLA3 GG genotype may be a limitation in this sub-group analyses.

---

## [Decision Letter · Decision Letter 1]

8 Oct 2019

Factors influencing subclinical atherosclerosis in patients with biopsy-proven nonalcoholic fatty liver disease

PONE-D-19-17091R1

Dear Dr. Atsukawa,

We are pleased to inform you that your manuscript has been judged scientifically suitable for publication and will be formally accepted for publication once it complies with all outstanding technical requirements.

With kind regards,

Jee-Fu Huang, M.D., Ph.D.

Academic Editor

PLOS ONE

Additional Editor Comments (optional):

Reviewers' comments:

Reviewer's Responses to Questions

**Comments to the Author**

1. If the authors have adequately addressed your comments raised in a previous round of review and you feel that this manuscript is now acceptable for publication, you may indicate that here to bypass the “Comments to the Author” section, enter your conflict of interest statement in the “Confidential to Editor” section, and submit your "Accept" recommendation.

Reviewer #1: All comments have been addressed

Reviewer #2: All comments have been addressed

Reviewer #3: All comments have been addressed

2. Is the manuscript technically sound, and do the data support the conclusions?

Reviewer #1: Yes

Reviewer #2: Yes

Reviewer #3: Partly

3. Has the statistical analysis been performed appropriately and rigorously? 

Reviewer #1: Yes

Reviewer #2: Yes

Reviewer #3: Yes

4. Have the authors made all data underlying the findings in their manuscript fully available?

Reviewer #1: Yes

Reviewer #2: Yes

Reviewer #3: Yes

5. Is the manuscript presented in an intelligible fashion and written in standard English?

Reviewer #1: Yes

Reviewer #2: Yes

Reviewer #3: Yes

6. Review Comments to the Author

Reviewer #1: The authors had addressed all issues well. I think the manuscripts get better after the revise. Thank you for the efforts of the authors.

Reviewer #2: The authors have satisfactorily responded to all my questions and made the necessary changes to the revised manuscript.

Reviewer #3: The previously raised issues have been addressed by the authors appropriately and I have no further comments.

7. PLOS authors have the option to publish the peer review history of their article (what does this mean?). If published, this will include your full peer review and any attached files.

Reviewer #1: No

Reviewer #2: No

Reviewer #3: No

---

## [Editor Report · Acceptance letter]

6 Nov 2019

PONE-D-19-17091R1 

Factors influencing subclinical atherosclerosis in patients with biopsy-proven nonalcoholic fatty liver disease 

Dear Dr. Atsukawa:

I am pleased to inform you that your manuscript has been deemed suitable for publication in PLOS ONE. Congratulations! Your manuscript is now with our production department. 

With kind regards,

on behalf of

Dr. Jee-Fu Huang 

Academic Editor

PLOS ONE